# Incidence and Predicting Factors of Histopathological Features at Robot-Assisted Radical Prostatectomy in the mpMRI Era: Results of a Single Tertiary Referral Center

**DOI:** 10.3390/medicina59030625

**Published:** 2023-03-21

**Authors:** Ernesto Di Mauro, Francesco Di Bello, Gianluigi Califano, Simone Morra, Massimiliano Creta, Giuseppe Celentano, Marco Abate, Agostino Fraia, Gabriele Pezone, Claudio Marino, Simone Cilio, Marco Capece, Roberto La Rocca, Ciro Imbimbo, Nicola Longo, Claudia Colla’ Ruvolo

**Affiliations:** Department of Neurosciences, Reproductive Sciences and Odontostomatology, University of Naples “Federico II”, Via Sergio Pansini 5, 80131 Naples, Italy

**Keywords:** upgrading, cribriform variant, active surveillance, perineural invasion, tumor length

## Abstract

*Background and Objectives*: To describe the predictors of cribriform variant status and perineural invasion (PNI) in robot-assisted radical prostatectomy (RARP) histology. To define the rates of upgrading between biopsy specimens and final histology and their possible predictive factors in prostate cancer (PCa) patients undergoing RARP. *Material and Methods:* Within our institutional database, 265 PCa patients who underwent prostate biopsies and consecutive RARP at our center were enrolled (2018–2022). In the overall population, two independent multivariable logistic regression models (LRMs) predicting the presence of PNI or cribriform variant status at RARP were performed. In low- and intermediate-risk PCa patients according to D’Amico risk classification, three independent multivariable LRMs were fitted to predict upgrading. *Results*: Of all, 30.9% were low-risk, 18.9% were intermediate-risk and 50.2% were high-risk PCa patients. In the overall population, the rates of the cribriform variant and PNI at RARP were 55.8% and 71.1%, respectively. After multivariable LRMs predicting PNI, total tumor length in biopsy cores (>24 mm [OR: 2.37, *p*-value = 0.03], relative to <24 mm) was an independent predictor. After multivariable LRMs predicting cribriform variant status, PIRADS (3 [OR:15.37], 4 [OR: 13.57] or 5 [OR: 16.51] relative to PIRADS 2, all *p* = 0.01) and total tumor length in biopsy cores (>24 mm [OR: 2.47, *p* = 0.01], relative to <24 mm) were independent predicting factors. In low- and intermediate-risk PCa patients, the rate of upgrading was 74.4% and 78.0%, respectively. After multivariable LRMs predicting upgrading, PIRADS (PIRADS 3 [OR: 7.01], 4 [OR: 16.98] or 5 [OR: 20.96] relative to PIRADS 2, all *p* = 0.01) was an independent predicting factor. *Conclusions*: RARP represents a tailored and risk-adapted treatment strategy for PCa patients. The indication of RP progressively migrates to high-risk PCa after a pre-operative assessment. Specifically, the PIRADS score at mpMRI should guide the decision-making process of urologists for PCa patients.

## 1. Introduction

Prostate cancer (PCa) is the second most frequent cancer in the male population, with an estimated 1.1 million diagnoses worldwide in 2012 [1,2]. The highest risk of PCa is in men over 50 years of age or age > 45 years with a family history of PCa [3,4,5,6]. The diagnosis is made by prostate biopsy in patients with abnormalities detected in their prostate-specific antigen (PSA) levels or digital rectal exam (DRE) [7,8,9]. In the last decade, there has been growth in the use of multiparametric magnetic resonance imaging (mpMRI) for the diagnosis and characterization of PCa based on the Prostate Imaging Reporting And Data System (PI-RADS v.2) [10,11].

According to the European Association of Urology (EAU) guidelines, histological grading of adenocarcinoma of the prostate is based on the definition of the International Society of Urological Pathology (ISUP) or Gleason grade group (GGG) modified grading, defined as ISUP 1 (Gleason score [GS] 6 [3 + 3]), ISUP 2 (GS 7 [3 + 4]), ISUP 3 (GS 7 [4 + 3]), ISUP 4 (GS 8 [4 + 4]) and ISUP 5 (GS 9 or 10) [2,8,9]. The GGG is the most important prognostic indicator for PCa [12,13,14,15]. Subsequent therapeutic decisions are based on the GGG, and it is able to predict the response to surgical treatment [12,13,16]. Other than the GGG, histological PCa variants are important clinicopathologic prognostic factors [1,17,18]. For example, many studies showed that cribriform glands correlated best with Gleason pattern (GP) 4 [19,20]. Cribriform glands are defined as a proliferation with multiple punched-out lumina, without intervening stroma [19,20]. One of the diagnostic criteria for its diagnosis is solid cribriform growth with more than 70% epithelial components in a gland surrounded by basal cells [20]. In 2014, the ISUP recommended that all cribriform glands, without exceptions, be graded as GP 4 [13]. The cribriform variant at radical prostatectomy (RP) specimen is correlated with reduced PCa-specific survival and increased risk of lymph node metastases [19,20]. Furthermore, another important histological factor has to be considered, namely perineural invasion (PNI), which is defined as cancer cell invasion in, around and through the nerves [21,22,23,24]. Historically, the complex neoplastic process of nerve invasion in PCa has been acknowledged as a significant route for the metastatic spread, independently from lymphatic or vascular involvement [21,24]. Indeed, PNI represents an unfavorable clinical and pathological characteristic of PCa, which has been associated with adverse features at final histology, biochemical recurrence (BCR), metastasis and reduction in overall survival [23,24,25,26,27]. The above features allowed patients to be placed in optimal risk categories. The EAU risk classification is based on the D’Amico Prostate Cancer Classification System, which categorizes groups of patients with a similar risk of BCR after surgery or radiotherapy (RT) by evaluating PSA, GGG at biopsy specimen and clinical T-stage [1,28,29,30]. Specifically, low-risk patients were defined as PSA < 10 ng/mL and T1–T2a and GGG 1, intermediate-risk patients were defined as PSA 10–20 ng/mL or T2b or GGG 2–3 and high-risk patients were defined as PSA > 20 ng/mL or T2c–T4 or GGG 4–5 [1,30]. Despite the referral to the D’Amico risk classification system, significant numbers of PCa patients experience tumor upgrading and/or upstaging between prostate biopsy and RP specimens in clinical practice [1,16,29,30,31,32,33,34]. As a result, the inclusion in active surveillance (AS) protocols for the low-risk PCa patient category may defer the final curative treatment for clinically localized PCa, affecting the PCa patients’ survival.

The primary aim of this study is to describe the independent predictors of cribriform variant status and PNI at final pathology in the overall population. The second aim is to define the rates of upgrading, upstaging and upgrading and/or upstaging between biopsy and surgery and their predictive factors in low- and intermediate-risk PCa patients treated with robotic-assisted RP (RARP) at our tertiary academic referral center.

## 2. Materials and Methods

### 2.1. Study Population

Within our retrospective PCa database, we identified PCa patients treated with RARP at our Tertiary Academic Referred Center from 2019 to 2022. Only patients who underwent prostate biopsies at our center were enrolled, obtaining the same pathological evaluation for both biopsy and final pathology specimens and avoiding the inter-observer variability. All the PCa patients included were clinically non-metastatic (cN0M0), based on MRI, total body computed tomography (CT) scan and bone scan and positron emission tomography (PET), with co-registration CT scan if not performed previously. PCa patients were stratified according to the D’Amico risk classification defined by the EAU guidelines as follows: low risk was defined as PSA < 10 ng/mL and T1–T2a and GGG 1, intermediate risk was defined as PSA 10–20 ng/mL or T2b or GGG 2–3 and high risk was defined as PSA > 20 ng/mL or T2c–T4 or GGG 4–5.

### 2.2. Variable Definition

For each patient enrolled, the following variables were recorded: age at diagnosis (continuously coded), PSA at diagnosis (ng/mL), PSA density, prostate volume (ml, based on MRI and if not available based on transrectal ultrasound), clinical T-stage (T1a, T1b, T1c, T2, T3) and PIRADS at prostate mpMRI. Furthermore, the following prostate biopsy data were recorded: total number of biopsy cores, number of positive biopsy cores, biopsy cores ratio (defined as the number of positive biopsy cores divided by the total number of biopsy cores), total tumor length at biopsy cores (defined as ≤24 mm or >24 mm, according to the overall median value) and GGG. Finally, pathological characteristics after surgery were collected: GGG, pathological T stage (T2a, T2b, T2c, T3a, T3b, T3c, T4), pathological N stage (N0, N1, Nx), presence of the cribriform variant (yes or no) and presence of perineural invasion (yes or no).

### 2.3. Statistical Analysis

Descriptive statistics were presented as medians and interquartile ranges (IQR) for continuously coded variables or counts and percentages for categorically coded variables. ANOVA and the Kruskal–Wallis test examined the statistical significance of ‘medians’, means and distributions differences. The chi-square test tested the statistical significance in the proportions’ differences. Two sets of analyses were performed: first, in the overall population, two independent univariable and multivariable logistic regression models (LRMs) were fitted to predict the presence of perineural invasion or cribriform variant status in the surgical specimen; second, in low- and intermediate-risk PCa patients, three independent LRMs were fitted to predict the rate of upgrading, defined as an increase of one or more GGG units at RARP specimen compared with biopsy, rate of upstaging defined as pT3 or higher (pT3+) and/or pN1 stages, and rate of upgrading and/or upstaging rate. In all statistical analyses, the R software environment for statistical computing and graphics (R version 3.6.1) was used. All tests were two-sided with the level of significance set at *p* < 0.05.

## 3. Results

### 3.1. Main Study Population

According to the inclusion criteria, 265 PCa patients who underwent RARP at our center were included. Of those, 82 (30,9%) were low risk, 50 (18.9%) were intermediate risk and 133 (50.2%) were high risk according to D’Amico risk classification (Table 1). The overall median age was 66 (IQR: 61–70, 65 [IQR: 61–69] vs. 68 [IQR: 65–71] vs. 66 [IQR: 60–70] in low-risk vs. intermediate-risk vs. high-risk PCa patients, respectively, *p*-value = 0.1). The overall median PSA at diagnosis was 6.6 (IQR: 4.9–9.4, 5.9 [IQR: 4.7–7.1] vs. 8.9 [IQR: 6.1–10.4] vs. 17 [IQR: 14.7–25.0] in low-risk vs. intermediate-risk vs. high-risk PCA patients, respectively, *p*-value < 0.001). Overall median PSA density was 0.2 (IQR: 0.1–0.3, 0.1 [IQR: 0.1–0.2] vs. 0.2 [IQR: 0.1–0.3] vs. 0.3 [IQR: 0.1–0.5] in low-risk vs. intermediate-risk vs. high-risk PCa patients, respectively, *p*-value < 0.001). The overall median prostate volume (ml) was 45.6 (IQR: 34–60). According to clinical T-stage, 220 (83%) patients were T2 (65 [80.5%] low-risk vs. 44 [88%] intermediate-risk vs. 110 [82.7%] high-risk PCa patients, respectively, *p*-value < 0.001). Moreover, 106 (40%) patients were PIRADS 4; of those, 30 (36.6%) were low-risk, 24 (48%) were intermediate-risk and 52 (39.1%) were high-risk PCa patients (*p*-value = 0.06). The overall median total number of biopsy cores was 16 (IQR: 16–16, 16 [IQR: 14.5–16] vs. 16 [IQR: 16–16] vs. 16 [IQR: 16–16] in low-risk vs. intermediate-risk vs. high-risk PCa patients, respectively, *p*-value = 0.06). The overall median number of positive biopsy cores was 5 (IQR: 3–8; 3 [IQR: 2–5] vs. 5 [IQR: 3–7] vs. 7 [IQR: 5–10] in low-risk vs. intermediate-risk vs. high-risk PCa patients, respectively, *p*-value < 0.001). The overall median cores ratio was 35.3 (IQR 18.8–56.2, 25 [IQR 12.5–35.6] vs. 31.2 [IQR 22.3–48.4] vs. 43.8 [IQR:31.2–62.5] in low-risk vs. intermediate-risk vs. high-risk PCa patients, respectively, *p*-value < 0.001). Off all, 136 (53.1%) PCa patients harbored total tumor length at biopsy cores ≤24 mm (65 [79.3%] low-risk vs. 27 [54.0%] intermediate-risk vs. 40 [30.1%] high-risk PCa patients, respectively) and 129 (48.7%) harbored total tumor length at biopsy cores >24 mm (16 [19.5%] low-risk vs. 21 [42%] intermediate-risk vs. 42 [69.2%] high-risk PCa patients respectively, *p*-value < 0.001). According to GGG at biopsy, 100 (37.7%) patients were GGG 1, 22 (8.3%) were GGG 2, 17 (6.4%) were GGG 3, 107 (40.4%) were GGG 4 and 19 (7.2%) were GGG 5.

Pathological findings after surgery are described in Table 2. According to the GGG at RARP specimen, 56 (21.1%) patients were GGG 1 (21 [25.6%] vs. 6 [12.0%] vs. 29 [21.8%] in low-risk vs. intermediate-risk vs. high-risk PCa patients, respectively), 17 (6.4%) were GGG 2 (6 [7.3%] vs. 3 [6.0%] vs. 8 [6.0%], in low-risk vs. intermediate risk vs. high-risk patients PCa, respectively), 21 (7.9%) were GGG 3 (10 [12.2%] vs. 2 [4.0%] vs. 9 [6.8%] in low-risk vs. intermediate-risk vs. high-risk PCa patients, respectively), 141 (53.2%) were GGG 4 (141 [53.2%] vs. 41 [50.0%] vs. 30 [60.0%] vs. 70 [52.6%] in low-risk vs. intermediate-risk vs. high-risk PCa patients, respectively), and 30 (11.3%) were GGG 5 (4 [4.9%] vs. 9 [18.0%] vs. 17 [12.8%] in low-risk vs. intermediate-risk vs. high-risk PCa patients, respectively). According to pathological T-stage, 136 (51.3%) patients were T2c (46 [56.1%] vs. 19 [38.0%] vs. 17 [12.8%] in low-risk vs. intermediate-risk vs. high-risk patients PCa, respectively, *p*-value = 0.04). According to pathological N-stage, 77 (29%) patients were N0, 0% low-risk vs. 6 (12%) intermediate-risk vs. 71 (53.4%) high-risk PCa patients, respectively (*p*-value = 0.04). The presence of perineural invasion at RARP specimen was recorded in 191 (72.1%) patients (55 [66.7%] vs. 40 [80.0%] vs. 96 [72.2%] were low-risk vs. intermediate-risk vs. high-risk PCa patients, respectively, *p*-value = 0.3). The presence of the cribriform variant at RARP specimen was recorded in 148 (55.8%) patients (44 [53.7%] vs. 29 [58.0%] vs. 75 [56.4%] were low-risk vs. intermediate-risk vs. high-risk PCa patients, respectively, *p*-value = 0.9).

### 3.2. Upgrading, Upstaging and Upgrading and/or Upstaging in Low-Risk and Intermediate-Risk PCa Patients

In low-risk PCa patients (82 [30.9%]), the rate of upgrading was 61 (74%). Of those, 6 (9.8%) upgraded to GGG 2, 10 (16.3%) upgraded to GGG 3, 41 (67.2%) upgraded to GGG 4, and 4 (6.5%) upgraded to GGG 5. The rate of upstaging was 72 (87.8%), and the rate of upgrading and/or upstaging was 76 (92.7%).

In intermediate-risk PCa patients (50 [18.9%]), the rate of upgrading was 39 (78.0%). Of those, 30 (76.9%) upgraded to GGG 4 and 9 (23.1%) upgraded to GGG 5. The rate of upstaging was 34 (68.0%), and the rate of upgrading and/or upstaging was 46 (92.0%) (Table 3).

### 3.3. Multivariable Logistic Regression Models Predicting Cribriform Variant status or Perineural Invasion in the Overall Population

After multivariable LRMs predicting the presence of the cribriform variant at RARP, PIRADS (PIRADS 3 [OR:15.37, 95% IC: 2.38–305.52, *p*-value = 0.01], PIRADS 4 [OR: 13.57, 95% IC 2.35–257.78, *p*-value = 0.01] or PIRADS 5 [OR: 16.51, 95% IC: 2.67–322.05, *p*-value = 0.01] relative to PIRADS 2) and total tumor length in biopsy cores (>24 mm [OR: 2.47, 95% IC: 1.17–2.50, *p*-value = 0.01], relative to ≤24 mm) were independent predicting factors. Conversely, age at diagnosis, PSA density, D’Amico risk classification, total tumor length in biopsy cores and number of positive biopsy cores were not independent predicting factors for the presence of the cribriform variant.

After multivariable LRMs predicting the presence of perineural invasion at RARP, total tumor length in biopsy cores (>24 mm [OR: 2.37, 95% IC: 1.06–5.41, relative to ≤24 mm, *p*-value = 0.03]) was an independent predicting factor. Conversely, age at diagnosis, PSA density, PIRADS, D’Amico risk classification, total number of biopsy cores, and the number of positive biopsy cores, were not independent predicting factors of the presence of perineural invasion (Table 4).

### 3.4. Multivariable Logistic Regression Models Predicting Upgrading, Upstaging, or Upgrading and/or Upstaging in Low and Intermediate-Risk PCa Patients

After multivariable LRMs predicting upgrading, PIRADS (PIRADS 3 [OR: 7.01, 95% IC: 0.79–159.18, *p*-value = 0.01], PIRADS 4 [OR: 16.98, 95% IC: 2.36–347.36, *p*-value = 0.01] or PIRADS 5 [OR: 20.96, 95% IC: 2.41–477.56, *p*-value = 0.01] relative to PIRADS 2) was an independent predicting factor. Conversely, age at diagnosis, PSA density, D’Amico risk classification, total number of biopsy cores, number of positive biopsy cores and total tumor length in biopsy cores were not independent predicting factors for upgrading.

After multivariable LRMs predicting upstaging, D’Amico risk classification (intermediate risk [OR: 3.33, 95% IC: 1.09–10.57, *p*-value = 0.03 relative to low risk] was an independent predicting factor. Conversely, age at diagnosis, PSA density, PIRADS, total number of biopsy cores, number of positive biopsy cores and total tumor length in biopsy cores were not independent predicting factors for upstaging.

After multivariable LRMs predicting upgrading and/or upstaging, no variables (age at diagnosis, PSA density, PIRADS, D’Amico risk classification, total number of biopsy cores, number of positive biopsy cores and total tumor length in biopsy cores) were independent predicting factors (Table 5).

## 4. Discussion

The final GGG represents the most important prognostic factor among all clinicopathologic features of PCa patients treated with RP [13,17,35]. Higher GGG on RP specimens is associated with advanced disease, higher rates of BCR, higher rates of metastases and higher risk of mortality [1,36]. Therefore, the preoperative GGG is a crucial variable included in PCa risk assessment tools, such as the D’Amico risk classification or cancer of the prostate risk assessment (CAPRA) score [28,29,30]. Nowadays, the absence of concordance between the biopsy and final histology reports represents one of the major concerns of PCa diagnosis and management due to the risk of mistreatment [37,38]. These aspects become particularly relevant in the context of AS, where a misinterpretation of GGG could defer the final curative treatment [39]. Moreover, other than GGG features at biopsy, the presence of histology variants is now recognized as an independent prognostic factor for PCa outcomes [40]. Therefore, the most recent PCa EAU guidelines suggest excluding men from AS when cribriform histology, perineural invasion or other histologic abnormalities (such as predominant intraductal carcinoma [IDC], sarcomatous, small cell carcinoma, extra prostatic extension or lymph vascular invasion) were also presented in needle biopsy specimens [1,19,26,41,42]. Recently, Flood et al. were enlightened that the presence of cribriform morphology on biopsy was strongly associated with upstaging and upgrading after RP [43]. Additionally, the authors also revealed that cribriform patterns have been associated with a larger tumor volume, extra prostatic extension (EPE), lymph node metastases and BCR following RP [44,45,46,47]. Similarly, Yu et al. assessed the role of the cribriform variant and IDC in addition to the CAPRA score in predicting BCR and death in 612 PCa patients treated with RP [48]. Specifically, they discovered that the CAPRA score outperformed the D’Amico classification at stratifying PCa patients in predicting BCR [48]. In aid of the prostate biopsy histological features, the role of mpMRI is constantly growing [49,50]. Much information can be obtained by MRI to predict surgical pathological findings, such as extracapsular extension (ECE), seminal vesicle invasion and node positivity [49,51,52]. Furthermore, mpMRI, specifically apparent diffusion coefficient (ADC) imaging reconstruction, has a high sensitivity for detecting cribriform patterns, especially when only one prostate lesion is identified [53]. However, according to the recent meta-analysis results assessing the high specificity but low sensitivity in predicting the pathological T-stage, the role of mpMRI remains debated, especially when compared with well-established risk classification models [49,54].

The aim of the current study was to describe the predictors of the cribriform variant and PNI at final histology in l PCa patients treated with RARP at our tertiary academic referral center. The second aim was to define the rates of upgrading, upstaging and upgrading and/or upstaging between biopsy specimens and final histology and their possible predictive factors in low- and intermediate-risk PCa patient cohorts. Our analysis denoted several noteworthy considerations.

First, half of the overall population treated with RARP belonged to the high-risk PCa category. This phenomenon is widely known as “PCa stage migration” and is consistent with previous authors’ results [55,56,57]. Indeed, Hoeh et al. showed the same phenomenon in 550 PCa patients prospectively collected at the Department of Urology of University Hospital Frankfurt, Germany [56]. The analysis demonstrated a significant decrease in the trend of low-risk and GS-6 PCa patients in contrast to an increasing trend of intermediate-risk/GS-8–10 PCa patients undergoing RP [56]. Moreover, Van Der Bergh et al. observed similar results in a multicenter study involving four European centers [58]. The authors highlighted a higher number of high-risk PCa patients treated with RP relative to the other risk categories [58]. This phenomenon might be explained by higher referral into the clinical practice of a tailored, risk-adapted treatment strategy without compromising functional and oncological safety but reducing the side effects for patients belonging to the lowest risk categories [56,58,59].

Second, in the overall population analysis, it emerged that almost two-thirds of patients harbored PNI at final pathology. This data agreed with previously published studies [23]. For example, Maru et al. recorded a prevalence of PNI of up to 75% for surgical resection specimens in 708 consecutive patients who underwent RP at The Methodist Hospital in Houston, Texas [23]. Moreover, we also observed that the total tumor length recorded in all biopsy cores was an independent predictor of PNI at RARP even after multivariable adjusting for clinical and pathological factors. To the best of our knowledge, no previous study examined the effect of biopsy tumor length on the presence of PNI at RARP. However, Brimo et al. evaluated 100 PCa patients who underwent RP in two tertiary referral centers in Montreal between the years 2000 and 2007 [60]. From their analysis, it emerged that there was a strong association between the extent of the cancer (as length in mm or as a percentage) and the BCR (*p* < 0.001) [60]. In conclusion, further studies should be conducted to corroborate or reject our findings.

Third, according to our results more than half of the overall population harbored cribriform variants at RARP. After multivariable adjustment, we recorded that the total tumor length at biopsy was an independent predictor of cribriform variant status at final histology. Specifically, we observed that a tumor length greater than 24 mm, which represented the median value of the current population, highly predicted the cribriform variant status. The high median value of 24 mm might be explained by the high number of biopsy cores (median value of 16 cores). Our observations could be justified by the amount of tumor obtained. The more tumor that was biopsied, the more chance the pathologist had to recognize a tumor variant histology. According to the most recent literature, a standard threshold of tumor length in the biopsy cores as a predictive factor for adverse pathology has not yet been assessed. For this reason, we chose the median as an objective value. Unfortunately, no previous studies examined the effect of tumor length on cribriform variant status at final pathology; in consequence, a comparison cannot be performed. A more robust result shown from our analyses was represented by the role of PIRADS at mpMR in predicting the cribriform variant. Specifically, PIRADS 3, 4 and 5 compared with PIRADS 2 had fifteen-, thirteen- and sixteen-fold higher risk to harbor cribriform variants at final pathology, respectively (*p* = 0.01). Our results agreed with Tuna et al.’s observations [53]. Indeed, the authors showed that the cribriform pattern was more frequently located in PIRADS 5 lesions [53]. Specifically, they noticed that cribriform-positive areas confirmed after prostate biopsy had ADC values lower than non-cribriform cancer areas within the primary index lesion [53]. Moreover, regardless of the cribriform histology pattern, Jyoti et al. demonstrated that the ADC ratio (calculated for each lesion by dividing the lowest ADC value in a lesion and the highest ADC value in normal prostate in peripheral zone) could discriminate GS-6 from GS-≥7 tumors at MRI imaging (*p* = 0.032) [61,62,63]. This observation was also enhanced by previous studies in which lower ADC values at prostate MRI imaging correlated with higher GGG at histological evaluation [62,63]. It is important to highlight that in 2014 the ISUP recommended that all cribriform glands, without exceptions, had to be graded as GP 4 [17]. In conclusion, MRI imaging (and specifically the ADC sequences) had the potential role to guide the biopsy needle into the most aggressive region of the PCa lesion and had high sensitivity in detecting clinically significant PCa foci and cribriform pattern, thereby improving the PCa patient risk assessment.

Fourth, we performed a subgroup analysis in low and intermediate PCa patients. From our report, it emerged that from 82 low-risk PCa patients, more than half upgraded or upstaged and more than 90% harbored one or both outcomes. Considering the GGG at final histology, two-thirds of low-risk PCa patients upgraded to GGG 4. Among the intermediate-risk PCa patients, more than half upgraded, two-thirds upstaged and 92% upgraded and/or upstaged. Considering GGG at final histology, more than two-thirds of intermediate-risk PCa patients upgraded to GGG 4. These data are corroborated by previous evidence. Porcaro et al. measured an upgrading rate at final pathology of 66.7% within 237 low-risk PCa patients who underwent RARP at a single tertiary referral center in Verona (vs. an upgrading rate of 74.4% in our population) [38]. Additionally, Zhao et al. examined 243 PCa patients diagnosed with standard biopsy or MRI-targeted biopsy [37]. Upgrading was observed in 100 and 75% of GGG 1 patients who underwent standard biopsy or MRI-targeted biopsy, respectively [28]. Moreover, Ahdoot et al. assessed valuable findings, comparing standard biopsy with MRI-targeted biopsy [64]. Indeed, the authors highlighted that the rate of any upgrading was statistically significantly higher for systematic biopsy (41.6%) and MRI-targeted biopsy (30.9%) than for combined biopsy (14.4%) [64]. However, differently from Zhao et al. and Ahdoot et al.’s studies, we did not stratify PCa patients into standard vs. targeted biopsy since over two-thirds [75.8%] of the overall sample received a cognitive biopsy based on the mpMRI findings with a median number of biopsy cores of 16. In conclusion, MRI-targeted biopsy alone does not decrease the risk of upgrading compared with the combination of both biopsy modalities [35,64,65]. However, future studies are needed.

Fifth, after multivariable LRMs predicting upgrading in low- and intermediate-risk PCa patients, the PIRADS was an independent predictor factor. Specifically, PIRADS 3, 4 and 5 compared with PIRADS 2 had seven-, sixteen- and twenty-fold higher risks of upgrading at final pathology, respectively (*p* = 0.01). Song et al.’s results agreed with our findings, demonstrating that PIRADS scores of 4 to 5 were highly associated with upgrading in 504 patients who underwent RP at their institution between 2011 and 2013 (OR: 2.26, CI: 1.46–3.50, *p* < 0.001) [66]. Furthermore, we also showed that when PIRADS is not available, the risk of upgrading in low- and intermediate-PCa patients was twenty-fold higher than in PIRADS 2 (*p* = 0.01). A possible explanation is due to the missing of a well-defined area target that could guide the TRUS biopsy.

Taken together, the GGG at biopsy guides the choice of the final curative treatment for PCa patients. Over the past few years, the indication of RP is migrating to high-risk more than low-risk PCa patients. The PCa stage migration ensures offering RP only to men who may have an oncological benefit with the treatment, avoiding unnecessary side effects in the others. Furthermore, cribriform histology or PNI should be identified to better assess the risk category to avoid mistreatment. In aid of the prostate biopsy histological features, the role of mpMRI is constantly growing. Indeed, PIRADS stratifications, and specifically ADC reconstructions of index lesions, might represent important data for tumor characterization. The MRI must be included in the risk classification systems available to guide the urologists during the PCa patients’ decision-making process.

Our analysis is not devoid of limitations. First, the retrospective nature of our study represents an inherent limitation. However, this is common with all similar observational studies. Second, our sample size of 265 PCa patients is relatively small. However, to reduce the inter-variability of the pathologist observations, we decided to only include patients who received prostate biopsy and RARP in our tertiary referral center. Third, considering that a European population was studied in the current analyses, we only referred to the EAU guidelines, excluding the National Comprehensive Cancer Network (NCCN) guidelines on PCa. Moreover, the PCa risk classification based on the NCCN guidelines considers the dimensions of the single core and the tumor invasion of the single core in prostate biopsy specimens, which are data not available in our pathological report. Fourth, we also included patients with PIRADS 2 at mpMRI. According to the EAU guidelines, we performed a prostate biopsy in this setting of patients only when the PSA, PSA density and digital rectal examination was highly indicative of prostate cancer. Fifth, the dataset does not include functional and oncological information on the RARP patient’s follow-up. Finally, we cannot exclude some residual bias that may have affected our results.

## 5. Conclusions

RP represents a tailored and risk-adapted treatment strategy for PCa. The indication of RP is progressively migrating from low- and intermediate-risk to high-risk PCa patients after a timely pre-operative assessment, which should include mpMRI findings. According to the most recent guidelines, an improvement of PCa risk classification systems is needed.

## Figures and Tables

**Table 1 medicina-59-00625-t001:** Baseline characteristics of 265 PCa patients stratified according to D’Amico risk classification. “Core ratio” is defined as the ratio between the number of positive biopsy cores and of obtained biopsy cores.

		Overall N = 265	Low-Risk PCa Patients N = 82 (30.9)	Intermediate-Risk PCa Patients N = 50 (18.9)	High-Risk PCa Patients N = 133 (50.2)	*p*-Value
Age (years)	Median (IQR)	66 (61–70)	65 (61–69)	68 (65–71)	66 (60–70)	0.1
PSA at diagnosis (ng/mL)	Median (IQR)	6.6 (4.9–9.4)	5.9 (4.7–7.1)	8.9 (6.1–10.4)	17 (14.7–25)	<0.001
PSA density	Median (IQR)	0.1 (0.1–0.2)	0.1 (0.1–0.2)	0.2 (0.1–0.3)	0.3 (0.1–0.5)	<0.001
Prostate volume (mL)	Median (IQR)	45.6 (34–60)	50 (42.2–62)	45 (30–60)	44 (34–55)	0.3
Clinical T-stage	1a	0 (0)	0 (0)	0 (0)	0 (0))	<0.001
	1b	8 (3)	5 (6.1)	1 (2)	2 (1.5)	
	1c	22 (8.3)	11 (13.4)	5 (10)	6 (4.5)	
	2	220 (83)	66 (80.5)	44 (88)	110 (82.7)	
	3	15 (5.7)	0 (0)	0 (0)	15 (11.3)	
PIRADS	2	10 (3.8)	3 (3.7)	3 (6)	4 (3)	0.6
	3	35 (13.2)	9 (11)	5 (10)	21 (15.8)	
	4	106 (40)	30 (36.6)	24 (48)	52 (39.1)	
	5	50 (18.9)	14 (17.1)	9 (18)	27 (20.3)	
	N/A	64 (24.2)	26 (31.7)	9 (18)	29 (21.8)	
GGG at biopsy	1	100 (37.7)	82 (100)	13 (26)	5 (3.8)	<0.001
	2	22 (8.3)	0 (0)	21 (42)	1 (0.8)	
	3	17 (6.4)	0 (0)	16 (32)	1 (0.8)	
	4	107 (40.4)	0 (0)	0 (0)	107 (80.5)	
	5	19 (7.2)	0 (0)	0 (0)	19 (14.3)	
Total number of biopsy cores	Median (IQR)	16 (16–16)	16 (14.5–16)	16 (16–16)	16 (16–16)	0.06
Number of positive biopsy cores	Median (IQR)	5 (3–8)	3 (2–5)	5 (3–7)	7 (5–10)	<0.001
Cores ratio	Median (IQR)	35.3 (18.8–56.2)	25 (12.5–35.6)	31.2 (22.3–48.4)	43.8 (31.2–62.5)	<0.001
Total tumor length at biopsy cores (n,%)	≤24	136 (53.1)	65 (79.3)	27 (54)	40 (30.1)	<0.001
	>24	129 (48.7)	16 (19.5)	21 (42)	42 (69.2)	

Abbreviations: GGG: Gleason grade group; IQR: interquartile ranges; N/A: not applicable; PCa: prostate cancer, PIRADS: Prostate Imaging Reporting and Data System; PSA: prostate-specific antigen.

**Table 2 medicina-59-00625-t002:** Tumor characteristics of 265 PCa patients at RP specimen, stratified according to D’Amico risk classification.

		Overall N = 265	Low-Risk PCa Patients N = 82 (30.9)	Intermediate-Risk PCa Patients N = 50 (18.9)	High-Risk PCa Patients N = 133 (50.2)	*p*-Value
GGG at RP specimen	1	56 (21.1)	21 (25.6)	6 (12)	29 (21.8)	0.1
	2	17 (6.4)	6 (7.3)	3 (6)	8 (6)	
	3	21 (7.9)	10 (12.2)	2 (4)	9 (6.8)	
	4	141 (53.2)	41 (50)	30 (60)	70 (52.6)	
	5	30 (11.3)	4 (4.9)	9 (18)	17 (12.8)	
Pathological T-stage	2a	41 (15.5)	10 (12.2)	7 (14)	24 (18)	0.4
	2b	31 (11.7)	7 (8.5)	9 (18)	15 (11.3)	
	2c	136 (51.3)	46 (56.1)	19 (38)	71 (53.4)	
	3a	40 (15.1)	13 (15.9)	12 (24)	15 (11.3)	
	3b	15 (5.7)	5 (6.1)	3 (6)	7 (5.3)	
	3c	1 (0.4)	1 (1.2)	0 (0)	0 (0)	
	4	1 (0.4)	0 (0)	0 (0)	1 (0.8)	
Pathological N-stage	0	77 (29.0)	0 (0)	6 (12.0)	71 (53.4)	0.02
	1	83 (31.3)	0 (0)	25 (50.0)	58 (43.6)	
	x	105 (39.7)	82 (100.0)	19 (38.0)	4 (3.0)	
Presence of cribriform variant at RP specimen	no	117 (44.2)	38 (46.3)	21 (42)	58 (43.6)	0.9
	yes	148 (55.8)	44 (53.7)	29 (58)	75 (56.4)	
Presence of perineural invasion at RP specimen	no	74 (27.9)	27 (32.9)	10 (20)	37 (27.8)	0.3
	yes	191 (72.1)	55 (67.1)	40 (80)	96 (72.2)	

Abbreviations: GGG: Gleason grade group; PCa: prostate cancer; RP: radical prostatectomy.

**Table 3 medicina-59-00625-t003:** Rates of upgrading, upstaging and upgrading and/or upstaging of 132 PCa patients who underwent RARP, stratified according to D’Amico risk classification.

		Low-Risk PCa Patients N = 82	Intermediate-Risk PCa Patients N = 50
Upgrading	No	21 (25.4)	11 (22)
	Yes	61 (74.4)	39 (78)
GGG at RP specimen	1	-	-
	2	6 (9.8)	-
	3	10 (16.3)	-
	4	41 (67.2)	30 (76.9)
	5	4 (6.5)	9 (23.1)
Upgrade to 2 + 3		16 (26.2)	-
Upgrade to 4 + 5		45 (73.7)	39 (100)
Upgrading to 3 + 4 + 5		55 (90.1)	-
Upstaging	No	10 (12.2)	16 (32.0)
	Yes	72 (87.8)	34 (68.0)
Upgrading and/or upstaging	No	6 (7.3)	4 (8)
	Yes	76 (92.7)	46 (92)

Abbreviations: GGG: Gleason grade group; PCa: prostate cancer; RP: radical prostatectomy.

**Table 4 medicina-59-00625-t004:** Weighted multivariable-adjusted logistic regression analyses predicted the effect of cribriform variant and perineural invasion presence within D’Amico risk classification of PCa patients, respectively.

	Perineural Invasion		Cribriform Variant	
	OR (95% CI)	*p*-Value	OR (95% CI)	*p*-Value
Age	1.02 (0.97–1.07)	0.3	0.97 (0.93–1.01)	0.2
PSA density	2.71 (0.27–35.48)	0.4	0.28 (0.03–2.16)	0.2
PIRADS				
PIRADS 2	Ref.	-	Ref.	*-*
PIRADS 3	1.97 (0.40–9.1)	0.3	15.37 (2.38–305.52)	0.01
PIRADS 4	1.55 (0.36–6.08)	0.5	13.57 (2.35–257.78)	0.01
PIRADS 5	2.32 (0.49–10.17)	0.2	16.51 (2.67–322.05)	0.01
PIRADS N/A	1.70 (0.37–7.08)	0.4	7.77 (1.29–149.81)	0.06
D’amico risk classification				
Low risk	Ref.	-	Ref.	-
Intermediate risk	1.56 (0.64–3.99)	0.3	1.07 (0.46–2.50)	0.8
High risk	1.08 (0.53–2.18)	0.8	0.72 (0.34–1.49)	0.3
Total number of biopsy cores	1.00 (0.86–1.17)	0.9	1.07 (0.93–1.24)	0.2
Number of positive biopsy cores	0.93 (0.82–1.05)	0.2	1.00 (0.89–1.12)	0.9
Total tumor length in biopsy cores				
≤24	Ref.	-	Ref.	-
>24	2.37 (1.06–5.41)	0.03	2.47 (1.17–2.50)	0.01

Abbreviations: CI: confidence interval; GGG: Gleason grade group; N/A: not available; OR: odds ratio; PCa: prostate cancer; PIRADS: Prostate Imaging Reporting and Data System.

**Table 5 medicina-59-00625-t005:** Weighted multivariable-adjusted^1^ logistic regression analyses predicted the effect of upgrading, upstaging and upgrading and/or upstaging presence within D’Amico risk classification of PCa patients, respectively.

	Upgrading		Upstaging		Upgrading and/or Upstaging	
	OR (95% CI)	*p*-Value	OR (95% CI)	*p*-Value	*p*-Value	*p*-Value
Age	0.99 (0.92–1.07)	0.9	0.94 (0.85–1.02)	0.2	0.90 (0.77–1.04)	0.2
PSA density	0.62 (0.01–61.2)	0.8	0.33 (0.01–14.6)	0.5	15.51 (0.01–131,058.44)	0.5
PIRADS						
PIRADS 2	Ref.	-	Ref.	-	Ref.	-
PIRADS 3	7.01 (0.79–159.18)	0.1	1.28 (0.12–13.63)	0.8	2.84 (0.20–40.06)	0.4
PIRADS 4	16.98 (2.36–347.36)	0.01	4.36 (0.51–38.66)	0.1	11.82 (0.90–165.09)	0.05
PIRADS 5	20.96 (2.41–477.56)	0.01	4.56 (0.45–50.34)	0.1	11.58 (0.65–356.12)	0.1
PIRADS NA	20.29 (2.51–441.98)	0.01	7.91 (0.73–93.01)	0.08	7.17 (0.45–116.23)	0.1
Risk classification of D’Amico						
Low risk	Ref.	-	Ref.	-	Ref.	-
Intermediate risk	1.49 (0.51–4.69)	0.5	0.29 (0.09–0.91)	0.03	1.31 (0.2–10.73)	0.7
Total number of biopsy cores	0.92 (0.72–1.14)	0.4	0.96 (0.74–1.24)	0.8	0.94 (0.62–1.44)	0.8
Number of positive biopsy cores	0.88 (0.73–1.06)	0.2	0.98 (0.78–1.23)	0.8	0.78 (0.57–1.06)	0.1
Total tumor length in biopsy cores						
≤24	Ref.	-	Ref	-	Ref	-
>24	2.20 (0.61–8.81)	0.2	2.92 (0.68–14.51)	0.1	10.53 (1.15–270.54)	0.06

Abbreviations: CI: confidence interval; N/A: not available; OR: odds ratio; PCa: prostate cancer; PIRADS: Prostate Imaging Reporting and Data System.

## Data Availability

Data available on demand due to restrictions such as confidentiality or ethics.

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
