# Peer review of "Incidence and Predicting Factors of Histopathological Features at Robot-Assisted Radical Prostatectomy in the mpMRI Era: Results of a Single Tertiary Referral Center"

_medicina, 2023, doi:10.3390/medicina59030625_

Round 1

Reviewer 1 Report

Di Mauro et al. investigated the effect of predictors for worse outcomes at RP, relying on a tertian care database after initial prostate biopsy.

The authors should be congratulated for their made for. However, before considering the current manuscript for publication, I suggest to modify the following points:

1)  Authors should clarify the exams performed for the clinical evaluation of lymph nodes (cN) during staging

2) Are data available on the tumor invasion of the single core? Why did the Authors use only EAU guidelines for referral and not the NCCN guidelines?

3) Are data available on the oncological outcomes after RARP for the included patients?

A major revision is required.

Author Response

On behalf of all authors, I would like to thank the Editorial Board and the Reviewers for the constructive comments, that addressed the manuscript titled “Incidence and predicting factors of histopathological features at robot-assisted radical prostatectomy in the mpMRI era:  results of a single tertiary referral center”.

Below, we provide a point-by-point list of replies and corrections that were made within the manuscript. We hope that the corrected version of the manuscript will now be considered satisfactory for publication.

  1. Thank you for the pertinent comment. In the current analyses we only included clinical N0 and M0 patients after imaging evaluation as suggested by the most recent European urology association (EAU) guidelines. We better clarified this point in the materials and methods section of the new version of the manuscript, which now reads as follows: ”All the PCa patients included were clinically non-metastatic (cN0M0), based on MRI, total body computed tomography (CT) scan and bone scan and Positron Emission Tomography (PET), with co-registration CT scan if not performed before.”
  2. Thank you for the comment. Data concerning the tumor invasion of the single core was not available in the pathological report of our Institution. Thus, we preferred to refer only to the EAU risk classification rather than to the NCCN guidelines where a percent of tumor in a single core is a parameter used for risk classification. Moreover, a European population was used for the analysis, and consequently, we retained the European classification more appropriate to use. However, we updated the limitation section, which now reads as follows: “Our analysis is not devoid of limitations. First, the retrospective nature of our study represents an inherent limitation. However, this is common with all similar observational studies. Second, our sample size of 265 PCa patients is relatively small. However, to reduce the inter-variability of the pathologist observations, we decided to only include patients who received prostate biopsy and RARP in our tertiary referral center. Third, considering that a European population was studied in the current analyses, we only referred to the EAU guidelines, excluding the National Comprehensive Cancer Network (NCCN) guidelines on PCa. Moreover, the PCa risk classification based on the NCCN guideline considers the dimension of the single core and the tumor invasion of the single core at prostate biopsy specimens, which are data not available in our pathological report. Fourth, we also included patients with PIRADS 2 at mpMRI. Ac-cording to the EAU guidelines, we performed a prostate biopsy in this setting of patients only when the PSA, PSA density, and digital rectal examination was highly suspected of prostate cancer. Fifth, the dataset does not include functional and oncological information on the RARP patient’s follow-up. Finally, we cannot exclude some residual bias that may have affected our results.”
  3. Unfortunately, the database used for the current analysis did not include the oncological outcome data post RARP and that information cannot easily and quickly be collected. We hope to include them in further works. However, we added this point to the limitation section of the discussion, which now reads as follows: “Our analysis is not devoid of limitations. First, the retrospective nature of our study represents an inherent limitation. However, this is common with all similar observational studies. Second, our sample size of 265 PCa patients is relatively small. However, to reduce the inter-variability of the pathologist observations, we decided to only include patients who received prostate biopsy and RARP in our tertiary referral center. Third, considering that a European population was studied in the current analyses, we only referred to the EAU guidelines, excluding the National Comprehensive Cancer Network (NCCN) guidelines on PCa. Moreover, the PCa risk classification based on the NCCN guideline considers the dimension of the single core and the tumor invasion of the single core at prostate biopsy specimens, which are data not available in our pathological report. Fourth, we also included patients with PIRADS 2 at mpMRI. According to the EAU guidelines, we performed a prostate biopsy in this setting of patients only when the PSA, PSA density, and digital rectal examination was highly suspected of prostate cancer. Fifth, the dataset does not include functional and oncological information on the RARP patient’s follow-up. Finally, we cannot exclude some residual bias that may have affected our results.”

Reviewer 2 Report

This study highlights the predicting the presence of  of PNI and total tumor length in biopsy cores on cribriform variant, PI-RADS after Robotic assisted RP. Upgrading of risk group are also observed. This is an interesting reports. I have some comments and questions.

1.You have 16 T1a PC patients, how did they biopsied ? Because it is diagnosed by TURP.

2. Do you have institutional review board agreement when conducting this study? If you have, what the number was ?

3. We usually didnot perform biopsy when mpMRI reports that PI-RADS score below 3. How did you do the biopsy in what indications ?

Author Response

On behalf of all authors, I would like to thank the Editorial Board and the Reviewers for the constructive comments, that addressed the manuscript titled “Incidence and predicting factors of histopathological features at robot-assisted radical prostatectomy in the mpMRI era:  results of a single tertiary referral center”.

Below, we provide a point-by-point list of replies and corrections that were made within the manuscript. We hope that the corrected version of the manuscript will now be considered satisfactory for publication.

  1. We thank the Reviewer for the pertinent comment. Reading the clinical T1a reported in the manuscript, we noticed that the clinical T stage was mistyped. All the patients reported as T1a stage were T1b. We corrected this aspect in the revised manuscript and in the table.
  2. The agreement of the Institutional Review Committee on the retrospective nature of the study was not obtained. However, all patients who have undergone RARP at our institution sign a consent to the processing of personal data and the use of them for retrospective studies at the time of admission, as specified in the Informed Consent Statement at the end of the manuscript.
  3. We thank the reviewer for the interesting comment. According to the EAU guidelines, patients with high PSA density and with a suspect digital rectal examination, performed a prostate biopsy even if the mp MRI reported a PIRADS 2. This choice was even driven by the fact that not all the mpMRI were performed in our institution. Specifically, in our population, patients with PIRADS 2 at mp MRi were characterized by a median PSA density of 0.17 ng/mL (defined as “intermediate-high” according to PSA-density risk group) and a clinical T-stage of cT2. However, we included this point in the limitation section of the discussion, which now reads as follows: “Our analysis is not devoid of limitations. First, the retrospective nature of our study represents an inherent limitation. However, this is common with all similar observational studies. Second, our sample size of 265 PCa patients is relatively small. However, to reduce the inter-variability of the pathologist observations, we decided to only include patients who received prostate biopsy and RARP in our tertiary referral center. Third, considering that a European population was studied in the current analyses, we only referred to the EAU guidelines, excluding the National Comprehensive Cancer Network (NCCN) guidelines on PCa. Moreover, the PCa risk classification based on the NCCN guideline considers the dimension of the single core and the tumor invasion of the single core at prostate biopsy specimens, which are data not available in our pathological report. Fourth, we also included patients with PIRADS 2 at mpMRI. According to the EAU guidelines, we performed a prostate biopsy in this setting of patients only when the PSA, PSA density, and digital rectal examination was highly suspected of prostate cancer. Fifth, the dataset does not include functional and oncological information on the RARP patient’s follow-up. Finally, we cannot exclude some residual bias that may have affected our results.”

Round 2

Reviewer 1 Report

the authors addressed my concerns sufficiently.